# Artificial intelligence for the classification of fractures around the knee in adults according to the 2018 AO/OTA classification system

**Anna Lind, Ehsan Akbarian, Simon Olsson, Hans Nåsell, Olof Sköldenberg, Ali Sharif Razavian, Max Gordon** (ORCID) *

Department of Clinical Sciences, Danderyd Hospital, Karolinska Institutet, Stockholm, Sweden

* max.gordon@ki.se

## Abstract

### Background

Fractures around the knee joint are inherently complex in terms of treatment; complication rates are high, and they are difficult to diagnose on a plain radiograph. An automated way of classifying radiographic images could improve diagnostic accuracy and would enable production of uniformly classified records of fractures to be used in researching treatment strategies for different fracture types. Recently deep learning, a form of artificial intelligence (AI), has shown promising results for interpreting radiographs. In this study, we aim to evaluate how well an AI can classify knee fractures according to the detailed 2018 AO-OTA fracture classification system.

### Methods

We selected 6003 radiograph exams taken at Danderyd University Hospital between the years 2002–2016, and manually categorized them according to the AO/OTA classification system and by custom classifiers. We then trained a ResNet-based neural network on this data. We evaluated the performance against a test set of 600 exams. Two senior orthopedic surgeons had reviewed these exams independently where we settled exams with disagreement through a consensus session.

### Results

We captured a total of 49 nested fracture classes. Weighted mean AUC was 0.87 for proximal tibia fractures, 0.89 for patella fractures and 0.89 for distal femur fractures. Almost ¾ of AUC estimates were above 0.8, out of which more than half reached an AUC of 0.9 or above indicating excellent performance.

### Conclusion

Our study shows that neural networks can be used not only for fracture identification but also for more detailed classification of fractures around the knee joint.

**Data Availability Statement:** There are legal and ethical restrictions on sharing the full data set. After discussions with the legal department at the

Karolinska Institute we have decided that the double-reviewed test-set can be shared without violating EU-regulations. The deidentified share the test dataset with 600 images are available through the data-sharing platform provided by AIDA (https://datasets.aida.medtech4health.se/10.23698/aida/kf2020). The version used for training the network has been uploaded to GitHub, see https://github.com/AliRazavian/TU.

**Funding:** This project was supported by grants provided by Region Stockholm (ALF project), and by the Karolinska Institute. Salaries and computational resources were funded through these grants. DeepMed AB provided no financial or intellectual property for this study.

**Competing interests:** The authors have read the journal's policy and the authors of this manuscript have the following competing interests: MG, OS, and AS are co-founders and shareholders in DeepMed AB. While this puts us at a financial competing interest, the company has limited activity with currently no revenue stream, no external investors and no pending patents. There are no patents, products in development or marketed products to declare. This does not alter our adherence to PLOS ONE policies on sharing data and materials.

## Introduction

Fractures around the knee joint are inherently complex with high risk of complications. For instance, during the first decade after a tibial plateau facture 7% receive a total knee replacement, five times more than the control population [1]. Bicondylar tibia fractures have a hazard ratio of 1.5 for total knee replacement, while high age has hazard ratio of 1.03 [1]. While regular primary osteoarthritis replacements have a survival rate of at least 95% in a decade, posttraumatic knee replacements have both higher complication rates and survival rates as low as 80% for the same time period [2]. There is a need to lessen complications from these fractures, and a reliable diagnosis and description of the fracture is crucial for providing correct treatment from the onset.

Experienced radiologists with extended orthopedic training constitute a scarce resource in many hospitals, especially in the middle of the night. Fatigue, inexperience and lack of time when interpreting diagnostic images increases the risk of human error as a cause for misdiagnosis [3–6]. Use of computed tomography (CT) might improve accuracy, but this is not universally true [7] and CT is not as readily available as plain radiographs. We believe that computer aided interpretation of radiographs could be of use both in helping clinicians properly assess the initial fracture as well as in retrospectively reviewing a large amount of fractures to better understand the optimal treatment regime.

Recent studies have shown promising results in applying deep learning, also known as deep neural networks, a form of artificial intelligence [8], for image interpretation. In medicine, deep learning has notably been explored in specialties such as endocrinology for retinal photography [9], dermatology for recognizing cancerous lesions [10] and oncology for recognizing pulmonary nodules [11], as well as mammographic tumors [12]. In trauma orthopedics, the last four years have yielded several studies on deep learning for fracture recognition with very promising results [4, 13–15], yet its applications and limitations are still largely unexplored [16].

There are to our knowledge no studies applying deep learning for knee fractures and there are very few published studies on fracture classification [14, 17, 18]. The primary aim of this study was therefore to evaluate how well a neural network can classify knee fractures according to the detailed 2018 AO-OTA fracture classification [19].

## Patients and methods

The research was approved by ethical review committee (dnr: 2014/453-31) (The Swedish Ethical Review Authority).

### Study design and setting

The study is a validation study of a diagnostic method based on retrospectively collected radiographic examinations. These examinations were analyzed by a neural network for both presence and type of knee fracture. Knee fracture is defined in this study as any fracture to the proximal tibia, patella or distal femur.

### Data selection

We extracted radiograph series around the knee taken between the years 2002 and 2016 from Danderyd University Hospital's Picture Archiving and Communication System (PACS). Images along with corresponding radiologist reports were anonymized. Using the reports, we identified phrases that suggested fractures or certain fracture subtypes. We then selected random subsets of image series from both the images with phrases suggesting that there may be a

fracture and those without. This selection generated a bias towards fractures and certain fracture subtypes to reduce the risk of non-fracture cases dominating the training data and rarer fractures being missed.

Radiograph projections included were not standardized. Trauma protocols as well as non-trauma protocols were included. Diaphyseal femur and tibia/fibula protocols were included as these display the knee joint although not in the center of the image. For each patient we only included the initial knee exam within a 90-day period to avoid overestimating the network by including duplicate cases of the same fracture at different stages. Images of knee fractures on children were tagged for exclusion by the reviewer upon seeing open physes as these are classified differently and Danderyd University Hospital only admits patients that are 15 years or older. Image series where the quality was deemed too poor to discern fracture lines were also tagged for exclusion by the reviewer. All tagged exclusions were then validated by MG before removal from the dataset.

## Method of classification

In this method of machine learning the neural network identifies patterns in images. The network is fed both the input (the radiographic images) and the information of expected output label (classifications of the fractures) in order to establish a connection between the features of a fracture and corresponding category [8].

Prior to being fed to the network the exams along with radiologist's reports were labelled using a custom-built platform according to AO/OTA-class (v. 2018) by AL, SO, MG & EA. The AO/OTA classification system was chosen as it can be applied to all three segments of the knee joint [19] and because of its level of detail. The classification system has more than 60 classes of knee fractures, many of which are nested and interdependent, e.g. the A1.1 is a subset of both A and A1 [19]. Fractures were classified down to lowest discernable subgroup or qualifier. (See S1 File for details). We also created custom output categories such as displacement/no displacement and lateral/medial fracture as it is interesting to see how well the network can discern these qualities regardless of AO/OTA class.

## Data sets

The data was randomly split into three sets: test, training and validation. The split into sets was constructed so that the same patient seeking and receiving an x-ray of the knee joint on multiple occasions with a > 90-day separation could be included multiple times in the same set, but there was no patient overlap between the training, validation and test sets.

The test set consisted of 600 cases, which were classified by two senior orthopedic surgeons, MG, OS and EA, working independently. Any disagreement was dealt with by a joint reevaluation session until a consensus between the two surgeons was reached. Out of the 600 cases, 71 cases had disagreement regarding type of fracture (see S1 File for details). The test set then served as the ground truth that the final network was tested against. A minimum of 2 captured cases per class was required for that class to be included in the test set. All images contained at least an AP and a lateral view and had to have the knee joint represented.

During training two sets of images were used, the training set which the network learned from and a validation set for evaluating performance and tweaking network parameters. The validation set was prepared in the same way as the test set but by AL and SO, two 4th year medical students. The training set was labeled only once by either AL or SO. MG validated all images with fractures or by the students marked for revisit. Initially, images were randomly selected for classification and fed to the network i.e. passive learning. As the learning progressed cases were selected based on the networks output: 1) initially cases with high

probability of a class were selected to populate each category, and then 2) cases where the network is the most uncertain to define the border was used i.e. active learning [20]. Due to the number of classes available the category used for selection changed depending on which categories were poorly performing at that stage. During this process the predictions from the network were fed back into the labeling interface as an additional feedback loop to the reviewers so that the error modes became clearer and could be addressed. The reviewers were presented with probabilities in the form of continuous color scale and categories with probability over 60% were preselected by the interface.

## Neural network setup

We used a convolutional neural network that was a modification of a ResNet type. The network consisted of a 26-layer architecture with batch normalization for each convolutional layer and adaptive max pool (See Table 1 for structure). Each class had a single endpoint that was converted into a probability using a sigmoid function. We randomly initialized the network and trained using stochastic gradient descent.

The training was split into several sessions with different regularizes for controlling overfitting. Between each session we re-set the learning rate and trained according to Table 2. We trained the network initially with dropout without any noise. In subsequent sessions we applied regularizers such as white noise, auto-encoders [21], semi-supervised learning with teacher-student networks [22] and stochastic weighted averaging [23]. During training we

**Table 1. General network architecture.**

| Type | Blocks | Kernel size | Filters | Section |
|---|---|---|---|---|
| Convolutional | 1 | 5x5 | 32 | Core |
| Convolutional | 1 | 3x3 | 64 | Core |
| ResNet block | 4x2 | 3x3 | 64 | Core |
| ResNet block | 2x2 | 3x3 | 128 | Core |
| ResNet block | 2x2 | 3x3 | 256 | Core |
| ResNet block | 2x2 | 3x3 | 512 | Core |
| Image max | 1 | - | - | Pool |
| Convolutional | 1 | 1x1 | 72 | Classification |
| Fully connected | 1 | - | 4 | Classification |
| Fully connected | 1 | - | 4 | Classification |

All images were individually processed in the core section of the network and then merged at the pool stage using the adaptive max function. The final classification section was then used for generating the AO/OTA classes.

**Table 2. The training setup of the network.**

| Session | Epochs | Initial learning rate | Noise | Teacher-student pseudo labels | Autoencoder | SWA |
|---|---|---|---|---|---|---|
| Initialization | 70 | 0.025 | none | no | no | no |
| Noise | 80 | 0.025 | 5% | no | no | no |
| Teacher-student | 40 | 0.010 | 5% | yes | no | no |
| Autoencoder | 20 | 0.025 | 10% | no | yes | no |
| SWA | 20 x 5 | 0.010 | 5% | no | no | yes |

All sessions used standard drop-out in addition to the above.

alternated between similar task for other anatomical sites, e.g. our ankle fracture dataset [17], using additional 16 172 exams. During the teacher student session, we augmented the dataset with unlabeled exams using a ratio of 1:2 where the teacher network had access to the radiologist report in addition to the images. The learning rate was adjusted at each epoch and followed the cosine function.

## Input images

The network was presented with all available radiographs in each series. Each radiograph was automatically cropped to the active image area, i.e. any black border was removed, and the image was reduced to a maximum of 256 pixels. We then added padding to the rectangular image so that the network received a square format of 256 x 256 pixels.

## Outcome measures & statistical analysis

Network performance was measured using area under curve (AUC) as primary outcome measure and sensitivity, specificity and Youden J as secondary outcome measures. Proportion of correctly detected fractures was estimated using AUC—the area under a receiver operating curve (ROC)—which is a plot of true positive rate against the false positive rate and suggests the networks ability to sort the class from low to high likelihood. An AUC value of 1.0 signifies prediction that is always correct and a value of 0.5 is no better than random chance. There is no exact guide for how to interpret AUC values, but in general an AUC of <0.7 is considered poor, 0.7–0.8 is considered acceptable, 0.8–0.9 is considered good to excellent and $\geq 0.9$ is considered excellent or outstanding [24–26]. Youden Index (J) is a value also used in conjunction with the ROC curve, it is a summary of sensitivity and specificity. It has a range of 0 to 1 and is defined as [26]:

$$J = sensitivity + specificity - 1$$

As there are many categories, we also presented a weighted mean of each measure that included all the subclasses, e.g. A-types will not only include the A-type but also all available groups and the subgroups into one measure. The weighting was according to the number of positive cases as we wanted small categories that may perform well by chance to have less influence on the weighted mean, for AUC the calculation was:

$$AUC_{weighted} = \frac{\sum_{i=1}^{categories} AUC_i * n_i}{\sum_{i=1}^{categories} n_i}$$

Cohens kappa, a measure of interrater reliability [27], was used to measure the level of agreement between the two human reviewers assessing the test set, as differences in interpretation between human reviewers could be a confounder in fairly assessing the network.

We implemented integrated gradients [28] as a method to access which image features the network analyzed to arrive at its output, as this is not otherwise immediately accessible. Integrated gradients displays this information as a heatmap where the color red illustrates image features that contribute positively to a certain output i.e. fracture category and blue illustrate features that contribute against that output [28].

The network was implemented and trained using PyTorch (v. 1.4). Statistical analysis was performed using R (4.0.0). The research was approved by ethical review committee (dnr: 2014/453-31).

## Results

From 42 163 available knee examinations 6188 exams were classified for the training set and 605 for the test set. A total of 70 images were excluded during classification, a majority as they contained open physes, leaving the training set with 6003 exams from 5657 separate patients and the test set with 600 from 526 patients (see Fig 1). Out of these 6003 exams, 5700 were used for training with an average 4.5 radiographs per exam (ranging 2 to 9 radiographs) while the remaining 303 were used for evaluating network performance and tweaking network

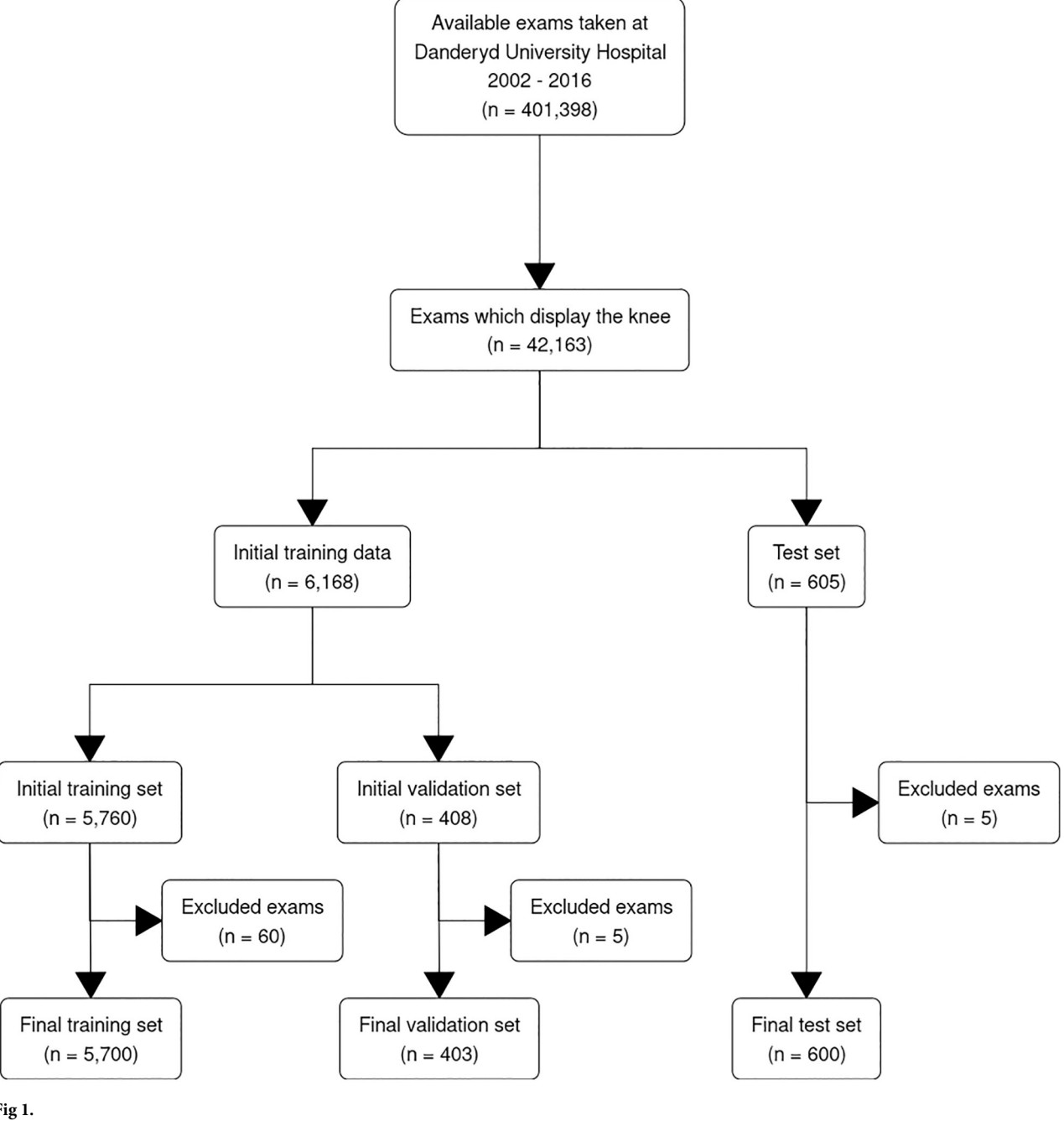

**Fig 1.**

parameters (the validation set). The test set had slightly fewer radiographs per exam, on average 4.1 (ranging from 2 to 7 radiographs). There was no patient overlap between the test and training datasets. We evaluated the network performance for a total of 49 fracture categories, 40 of which were AO/OTA classes and 9 custom classes.

### Proximal tibia (AO/OTA 41)—621 training cases and 68 evaluation cases

The weighted mean AUC for all tibial plateau fractures was 0.87 (95% CI, 0.82–0.92), sensitivity, specificity and Youden J were 0.83 (95% CI, 0.80–0.92), 0.91 (95% CI, 0.85–0.93) and 0.74 (95% CI, 0.69–0.83) respectively. As shown in Table 3, the A-types, which consisted mostly of tiny avulsions, performed the worst, around 0.7. B-types was closer to 0.9 and the C-types with subclasses just above 0.8. For the split-depression fractures (B3-group) performed excellently with all estimates above 0.9 Among the custom descriptors, medial and lateral performed with AUC scores of 0.89 and 0.81 respectively. The custom displacement class performed well with an AUC of 0.91.

### Patella (AO/OTA 34)—525 training cases and 40 evaluation cases

The weighted mean AUC for patella was 0.89 (95% CI, 0.83–0.94), sensitivity, specificity and Youden J were 0.89 (95% CI, 0.81–0.96), 0.88 (95% CI, 0.85–0.93) and 0.77 (95% CI, 0.70–0.87) respectively. Similar to proximal tibia fractures, the A-types (extraarticular fracture) had the lowest performance with AUC just under 0.8. The B-types, partial articular sagittal fractures, had the highest AUC-scores with around AUC 0.9 for the main group and all subgroups. The C-types, complete articular fractures, also performed well, only C1.3 (fractures in the distal third of the patella) performed below 0.8 (Table 4).

### Distal femur (AO/OTA 33)—147 training cases and 12 evaluation cases

Distal femur fractures were rare both in the training and the test data. Despite this, the weighted mean AUC was 0.89 (95% CI, 0.78–0.96), sensitivity, specificity and Youden J were 0.90 (95% CI, 0.82–1.00), 0.92 (95% CI, 0.79–0.97), and 0.81 (95% CI, 0.71–0.96) respectively. Only the B-type (partial articular fractures) performed lower at AUC 0.72. However, the number of cases were few and many of the confidence intervals were wide (Table 5).

### Inter-rater results

The Cohen's kappa between MG and EA ranged between 0 and 1 with a large variety between categories (see S2 Table in S1 File). High Cohen's kappa appeared to correspond weakly to classes where the network also performed well and there were indications that the number of training cases facilitated this effect (Fig 2). The correlation was however not strong enough to provide significant results using a linear regression.

### Network insight and example images

We sampled cases where the network was most certain of a prediction, whether correct or incorrect, for analysis. Case images for the most common fracture type in the data, proximal tibia B-type, and the adjacent C-type are shown below (Fig 3A to 3C). Also shown are heatmaps visualizing which areas in the images the network focuses on as colored dots. There were no clear discernable trends among these cases as to what made the network fail or succeed. Colored dots were concentrated to the joint segment of the bone and often seemed to cluster close to fracture lines, suggesting that the network appropriately finds these areas to contain relevant information.

**Table 3. Network performance for proximal tibia.**

| | Observed cases (n = 600) | Sensitivity (%) | Specificity (%) | Youden's J | AUC (95% CI) |
|---|---|---|---|---|---|
| **Proximal tibia** | | | | | |
| **A** | | | | | |
| A | 10 | 50 | 94 | 0.44 | 0.72 (0.52 to 0.91) |
| 1 | 8 | 60 | 82 | 0.42 | 0.73 (0.52 to 0.94) |
| ...3 | 5 | 80 | 79 | 0.59 | 0.78 (0.52 to 0.95) |
| ...→a | 3 | 100 | 76 | 0.76 | 0.86 (0.76 to 0.95) |
| A displaced | 3 | 67 | 93 | 0.60 | 0.87 (0.68 to 1.00) |
| **B** | | | | | |
| B | 47 | 83 | 88 | 0.71 | 0.89 (0.83 to 0.95) |
| 1 | 11 | 73 | 85 | 0.58 | 0.78 (0.60 to 0.91) |
| ...1 | 6 | 67 | 81 | 0.48 | 0.76 (0.59 to 0.91) |
| ...2 | 2 | 100 | 94 | 0.94 | 0.97 (0.93 to 1.00) |
| ...3 | 3 | 67 | 92 | 0.59 | 0.72 (0.24 to 1.00) |
| 2 | 10 | 67 | 90 | 0.57 | 0.74 (0.49 to 0.94) |
| ...1 | 6 | 83 | 93 | 0.76 | 0.89 (0.73 to 0.98) |
| ...2 | 4 | 100 | 81 | 0.81 | 0.88 (0.81 to 0.97) |
| 3 | 26 | 92 | 92 | 0.84 | 0.97 (0.95 to 0.99) |
| ...1 | 12 | 100 | 94 | 0.94 | 0.99 (0.97 to 0.99) |
| ...3 | 14 | 93 | 88 | 0.81 | 0.93 (0.85 to 0.98) |
| B → x | 5 | 100 | 93 | 0.93 | 0.97 (0.94 to 0.99) |
| B → t | 7 | 100 | 97 | 0.97 | 0.99 (0.97 to 0.99) |
| B → u | 6 | 83 | 93 | 0.76 | 0.88 (0.68 to 0.98) |
| **C** | | | | | |
| C | 11 | 82 | 95 | 0.77 | 0.83 (0.60 to 0.99) |
| 1 | 2 | 50 | 100 | 0.50 | 0.53 (0.06 to 1.00) |
| 2 | 4 | 100 | 98 | 0.98 | 0.99 (0.98 to 1.00) |
| 3 | 5 | 80 | 95 | 0.75 | 0.79 (0.43 to 0.98) |
| ...1 | 4 | 75 | 95 | 0.70 | 0.74 (0.30 to 0.98) |
| **Custom classes** | | | | | |
| Displaced | 29 | 83 | 97 | 0.80 | 0.91 (0.82 to 0.98) |
| Lateral | 14 | 75 | 90 | 0.65 | 0.81 (0.62 to 0.97) |
| Medial | 10 | 78 | 89 | 0.67 | 0.89 (0.74 to 0.98) |
| C2 or C3 | 5 | 80 | 94 | 0.74 | 0.80 (0.43 to 0.99) |
| Lateral B2 or B3 | 18 | 94 | 95 | 0.90 | 0.96 (0.88 to 0.99) |
| Medial B2 or B3 | 3 | 100 | 75 | 0.75 | 0.86 (0.76 to 0.96) |

Table showing network performance for the different AO-OTA classes as well as other fracture descriptors, first letter corresponds to fracture type, first number to group, second number to subgroup and last letter to qualifiers. The observed cases column correspond to the number of observed fractures by the reviewers. Note that an exam can appear several times as the category A1.3 will belong to both the overall A-type, A1 group and A1.3 subgroup at the same time.

## Discussion

This is, to our knowledge, the first study to evaluate a deep neural network for detailed knee fracture diagnostics. We evaluated a total of 49 fracture categories. In general, the network performed well with almost ¾ AUC estimates above 0.8. Out of these, a little more than half reached an AUC of 0.9 or above indicating excellent performance.

We conducted no direct comparison between network performance and performance of clinicians. Chung et al [14] in a similar study on deep learning for fracture classification found

**Table 4. Network performance for patella.**

| | Observed cases (n = 600) | Sensitivity (%) | Specificity (%) | Youden's J | AUC (95% CI) |
|---|---|---|---|---|---|
| | | | Patella | | |
| | | | A | | |
| A | 5 | 80 | 83 | 0.63 | 0.79 (0.67 to 0.86) |
| 1 | 5 | 80 | 85 | 0.65 | 0.81 (0.68 to 0.89) |
| 1a | 2 | 100 | 94 | 0.94 | 0.97 (0.93 to 0.99) |
| | | | B | | |
| B | 6 | 100 | 90 | 0.90 | 0.94 (0.91 to 0.97) |
| 1 | 6 | 100 | 90 | 0.90 | 0.93 (0.90 to 0.97) |
| ...1 | 3 | 100 | 78 | 0.78 | 0.86 (0.76 to 0.95) |
| ...2 | 3 | 100 | 89 | 0.89 | 0.95 (0.89 to 0.99) |
| | | | C | | |
| C | 29 | 90 | 86 | 0.75 | 0.90 (0.79 to 0.97) |
| 1 | 11 | 91 | 86 | 0.76 | 0.89 (0.74 to 0.97) |
| ...1 | 6 | 100 | 85 | 0.85 | 0.94 (0.89 to 0.99) |
| ...3 | 5 | 60 | 89 | 0.49 | 0.75 (0.44 to 0.96) |
| 2 | 8 | 100 | 88 | 0.88 | 0.97 (0.93 to 0.99) |
| 3 | 10 | 80 | 97 | 0.77 | 0.88 (0.70 to 0.98) |
| | | | Custom classes | | |
| Displaced | 21 | 81 | 91 | 0.72 | 0.88 (0.76 to 0.97 |

Table showing network performance for the different AO-OTA classes as well as other fracture descriptors, letter corresponds to fracture type, first number to group, second number to subgroup and last letter to qualifiers. The observed cases column correspond to the number of observed fractures by the reviewers. Note that an exam can appear several times as the category B1.1 will belong to both the overall B-type, B1 group and B1.1 subgroup at the same time.

that orthopedic surgeons specialized in shoulders performed with a Youden J, a summary of sensitivity and specificity, of 0.43–0.86 at classifying shoulder fractures. By that standard, our network performed with Youden J ranging from 0.42–0.98 and mean weighted Youden J 0.74–0.81 which would likely indicate similar results compared to orthopedic surgeons, with the caveat that fractures to the shoulder and knee might differ in diagnostic difficulty.

**Table 5. Network performance for distal femur.**

| | Observed cases (n = 600) | Sensitivity (%) | Specificity (%) | Youden's J | AUC (95% CI) |
|---|---|---|---|---|---|
| | | | Distal femur | | |
| | | | A | | |
| A | 5 | 100 | 83 | 0.83 | 0.94 (0.88 to 0.99) |
| 2 | 4 | 100 | 97 | 0.97 | 0.99 (0.97 to 1.00) |
| | | | B | | |
| B | 4 | 75 | 83 | 0.58 | 0.72 (0.31 to 0.96) |
| | | | C | | |
| C | 3 | 100 | 97 | 0.97 | 0.98 (0.97 to 1.00) |
| | | | Custom classes | | |
| B11 or C11 | 4 | 75 | 94 | 0.69 | 0.81 (0.49 to 0.98) |

Table showing network performance for the different AO-OTA classes as well as other fracture descriptors, letter corresponds to fracture type, first number to group and second number to subgroup. The observed cases column correspond to the number of observed fractures by the reviewers. Note that an exam can appear several times as the category A2 will belong to both the overall A-type, and A2 group at the same time.

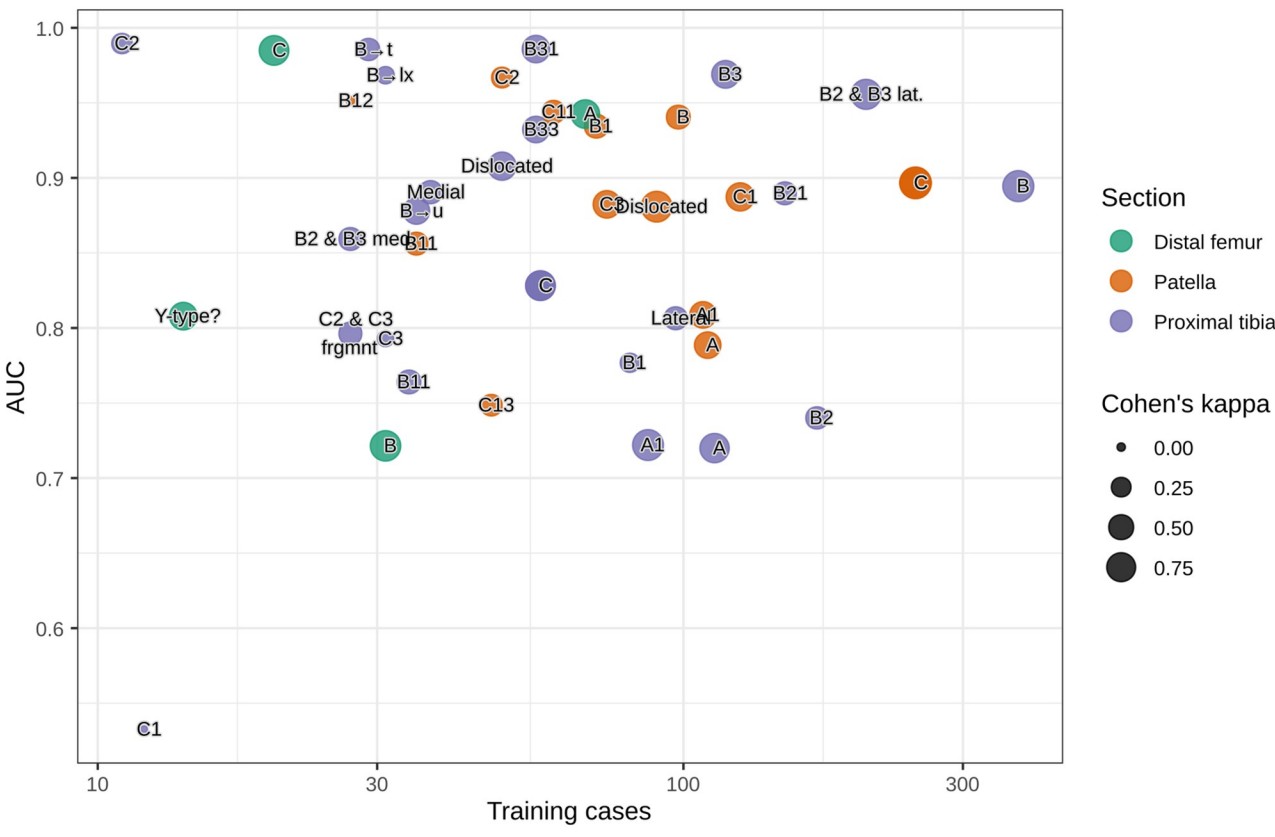

**Fig 2.**

Some fractures were classified with significantly better prediction than others, though in many cases differences in performance between categories were not significant. During training, we could see a trend where categories with few training cases performed worse, however this correlation diminished later on due to the active learning approach. There were also initially indications that fractures with low Cohen's kappa values were more challenging, but after re-visiting all fractures in the training set this effect was no longer detectable. The importance of reducing label noise i.e. disruptions which obscure the relationship between fracture characteristics and correct category [29],—sometimes stemming from incorrect or inconsistent labelling by the image reviewers—is well-established [30] and our experience aligns with prior findings.

Our diagnostic accuracy is somewhat lower than that reported in previous studies on deep learning for fracture diagnostics. Langerhuizen et al found in their 2019 systematic review [16] that six studies using a convolutional neural network to identify fractures on a plain radiograph [3, 4, 13, 14, 31, 32] reported AUC ranging from 0.95–1.0 and/or accuracy ranging from 83–97%. One of the studies in the review, Chung et al [14], also investigated fracture classification using a convoluted neural network, with an AUC of 0.90–0.98 depending on category. The difference in performance could partly be due to the complexity of the task at hand; our study had 49 nested fracture categories whereas Chung et al. [14] had 4. Another likely cause is that this study made use of a less strictly controlled environment in which to train and test the network. Four of the six mentioned studies in the systematic review only used one radiographic projection, [4, 14, 31, 32] a fifth study used two projections [3]. This study made use of

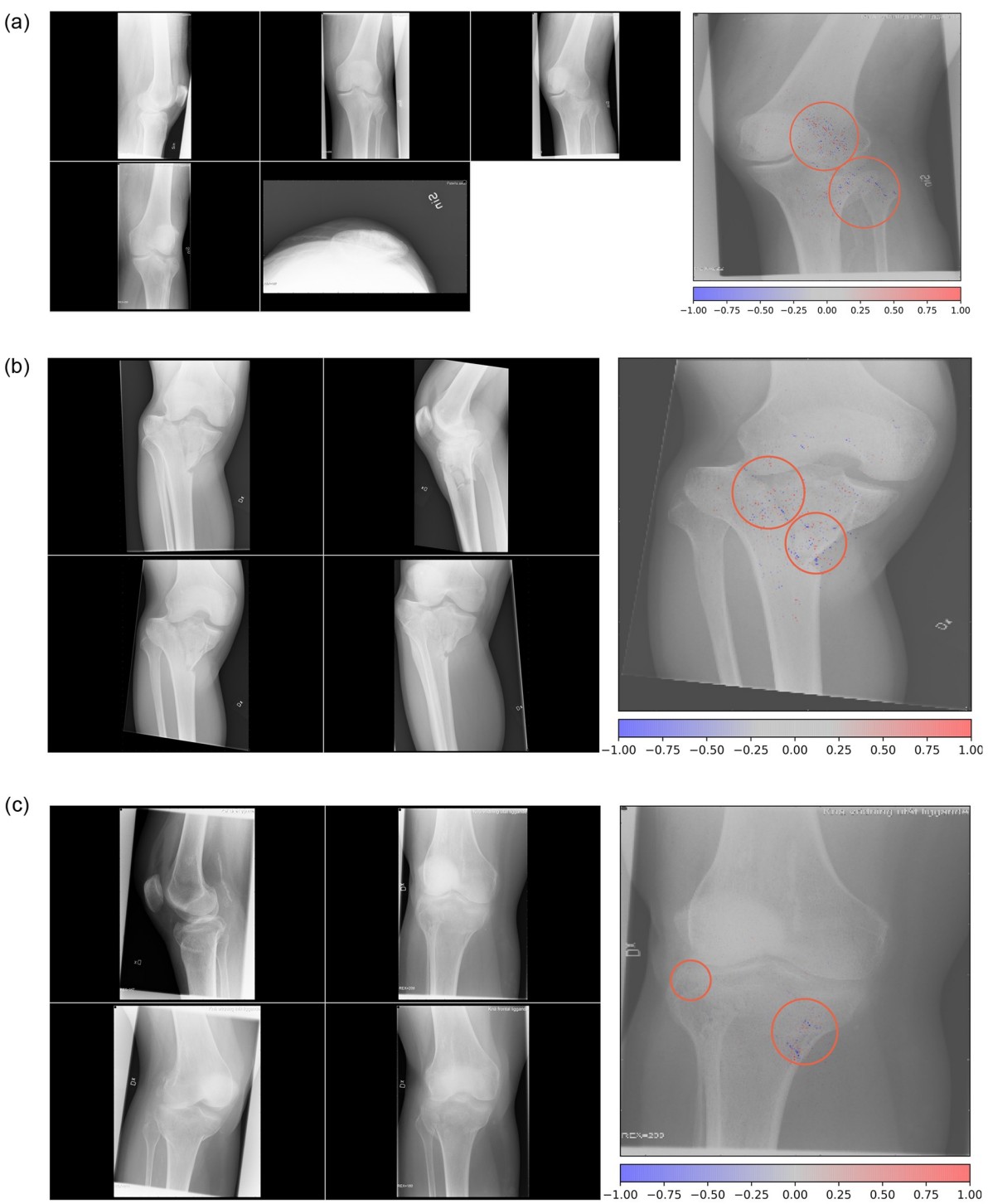

**Fig 3.**

several projections not all centered on the knee joint. Furthermore, our images where not centered around the fracture to the extent that images from the previous studies were and we did not remove images containing distracting elements such as implants, as Urakawa et al. did [31].

## Strengths and limitations

This study aimed to retain the full complexity a random influx of patients brings. We did not introduce selection bias by automatically excluding knees with contractures, implants, thick casts, and other visual challenges. Our study should thus be less likely to overestimate the AI by simplifying the diagnostic scenario and closer to achieving a clinically relevant setting as requested by Langerhuizen et al in their systematic review [16]. However, we did not avoid selection bias completely as we removed images where the image quality was to poor for the human reviewers to establish a correct fracture label. In the test set 5 cases where excluded, four due to open physes and one because it did not include the knee joint, see Fig 1. We actively selected rare fracture patterns, both to be able to capture all AO/OTA classes but also because we believe that, in the long run, the potential clinical value of a computer assisted diagnosis will not only be in everyday fractures but in rare cases where even the clinician is uncertain. This could however also be considered a limitation as we did introduce a bias towards having rare fractures overrepresented in our data compared to how often they appear in clinic. Fractures overall were also overrepresented as otherwise the data would be dominated by healthy images. This would present less challenge for the network and would likely yield the appearance of a better performing network but would hinder the goal of the study to evaluate network performance for classification of different fracture types. We believe that the mixed inter-rater agreement between the orthopedic surgeons reviewing the test sets also reflects that the network was evaluated on cases that would be of varying difficulty for clinicians instead of more trivial cases only.

A central limitation is that we did not have a sophisticated method of establishing ground truth labels such as utilizing CT/MRI scans or operative findings or other clinical data to aid the research team in interpreting the images. Including CT/MRI:s for 6000 exams was deemed unfeasible as this would have vastly increased the time to review each exam and something more suited for follow-up studies. Image annotation was instead aided by the radiologist report, written with access to patient history and other exams. Unfortunately, this report was often too simplistic to help in subgrouping AO/OTA-classes. Double audits were used for fracture images but there is still a risk of misclassification. This misclassification bias could have resulted in an underestimate in the number of complex fractures. However, we believe that fractures that may require surgery will be subjected to CT/MRI exams, even with the aid of computer-assisted diagnosis, as these are incredibly useful before entering the operating theatre.

The AO/OTA classification system leaves room for differences in interpretation between image reviewers—as demonstrated by Cohen kappa values between MG and EA—which likely impaired a completely fair judgement of network performance. The AO/OTA fracture classification system is also perhaps not the most commonly applied knee fracture classifier, as it is impractically extensive for many clinical settings. However, its level of detail can be useful for research purposes, and while some fractures where difficult to categorize, once we super-grouped many of the estimates we saw a significant boost, suggesting that this detailed classification can easily be simplified into one with fewer categories if need be.

While the fractures were collected from over a decade long period with a large sample of patients, our data selection was limited in that the data source is a single hospital in Stockholm. A fracture recognition tool developed from this network might not perform as well on the fracture panoramas of other cities or countries. Furthermore, findings are only applicable to an adult population.

### Clinical applications

The study evaluates a potential diagnostic tool with the ability to generate classifications or information which otherwise might fall into the area of knowledge for an orthopedic specialist rather than a radiologist. The AO-OTA classification carries relatively detailed information on properties usually not mentioned in the radiologist report, addition of a network report would provide extra information of value for the clinician treating the patient. This tool could also aid in alerting clinicians of otherwise potentially missed fissures and could serve as a built-in fail-safe or second opinion for clinicians.

### Future studies

Future studies could likely benefit from bringing in further information from medical records and x-ray referral and using more detailed imaging methods such as CT or MRI or operative findings as possible ways to refine the answer key the network is evaluated against. By using pre-training network as presented, it should be feasible to fine-tune the network using a more detailed but smaller subset of the cases used here.

In this study we relied on anonymized cases without patient data, adding patient outcomes can be of great interest as we usually want to connect the fracture pattern to the risk of complications. Having a computer aided diagnostic tools allows us to do this on an unprecedented scale.

## Conclusion

In conclusion, we found that a neural network can be taught to apply the 2018 AO/OTA fracture classification system to diagnose knee fractures with an accuracy ranging from acceptable to excellent for most fracture classes. It can also be taught to differ between medial and lateral fractures as well as non-displaced and displaced fractures. Our study shows that neural networks have potential not only for the task of fracture identification but for more detailed description and classification.

## Supporting information

**S1 File.**
(DOCX)

## Author Contributions

**Conceptualization:** Ehsan Akbarian, Olof Sköldenberg, Ali Sharif Razavian, Max Gordon.

**Data curation:** Anna Lind, Ehsan Akbarian, Simon Olsson, Hans Nåsell, Olof Sköldenberg, Max Gordon.

**Formal analysis:** Max Gordon.

**Funding acquisition:** Max Gordon.

**Investigation:** Anna Lind, Max Gordon.

**Methodology:** Anna Lind, Simon Olsson, Max Gordon.

**Project administration:** Max Gordon.

**Resources:** Max Gordon.

**Software:** Ali Sharif Razavian, Max Gordon.

**Supervision:** Max Gordon.

**Validation:** Hans Nåsell, Olof Sköldenberg, Max Gordon.

**Visualization:** Max Gordon.

**Writing – original draft:** Anna Lind.

**Writing – review & editing:** Anna Lind, Ehsan Akbarian, Simon Olsson, Hans Nåsell, Olof Sköldenberg, Ali Sharif Razavian, Max Gordon.

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
