## [Decision Letter · Decision Letter 0]

22 Oct 2020

PONE-D-20-26924

Artificial intelligence for the classification of knee fractures in adults according to the 2018 AO/OTA classification system

PLOS ONE

Dear Dr. Gordon,

Thank you for submitting your manuscript to PLOS ONE. After careful consideration, we feel that it has merit but does not fully meet PLOS ONE’s publication criteria as it currently stands. Therefore, we invite you to submit a revised version of the manuscript that addresses the points raised during the review process.

The reviewers are overall positive but they have also identified multiple major and minor issues that need to be addressed before the manuscript can be considered for publication. Especially, carefully update literature following comments of R1, provide detailed responses regarding the data description and experimental setup (R1, R2, R3), and provide details of the method following the comments of R3.

We look forward to receiving your revised manuscript.

Kind regards,

Ivana Isgum

Academic Editor

PLOS ONE

Journal Requirements:

2.We note that you have indicated that data from this study are available upon request. PLOS only allows data to be available upon request if there are legal or ethical restrictions on sharing data publicly. For information on unacceptable data access restrictions, please see http://journals.plos.org/plosone/s/data-availability#loc-unacceptable-data-access-restrictions.

3.Thank you for stating the following in the Competing Interests section:

[MG, OS and AS are co-founders and shareholders in DeepMed AB.].

4.Your ethics statement should only appear in the Methods section of your manuscript. If your ethics statement is written in any section besides the Methods, please move it to the Methods section and delete it from any other section. Please ensure that your ethics statement is included in your manuscript, as the ethics statement entered into the online submission form will not be published alongside your manuscript.

Reviewers' comments:

Reviewer's Responses to Questions

**Comments to the Author**

1. Is the manuscript technically sound, and do the data support the conclusions?

Reviewer #1: Partly

Reviewer #2: Yes

Reviewer #3: Yes

Reviewer #4: Yes

2. Has the statistical analysis been performed appropriately and rigorously? 

Reviewer #1: Yes

Reviewer #2: Yes

Reviewer #3: N/A

Reviewer #4: Yes

3. Have the authors made all data underlying the findings in their manuscript fully available?

Reviewer #1: Yes

Reviewer #2: Yes

Reviewer #3: No

Reviewer #4: Yes

4. Is the manuscript presented in an intelligible fashion and written in standard English?

Reviewer #1: Yes

Reviewer #2: Yes

Reviewer #3: Yes

Reviewer #4: Yes

5. Review Comments to the Author

Reviewer #1: This is an interesting article but requires some significant clarifications.

The knee is a joint, thus it dislocates but does not fracture. Please use a different word or indicate the bones involved.

Please clarify your sample sizes; I had great trouble sorting this out.

Please provide the number of independent patients -with the number of radiographs per patient.

Did each patient have all the views -did every plateau fracture have an evaluation with a paired AP and Lateral

i am not fully clear on machine learning technology - were normal x-rays included if the patient had a distal femur fracture?

I find the fact that you quote that knee replacements only last a decade in 80% of the population fairly unrealistic - please review the literature to be sure your quoted reference reflects the global feeling.

In the United States very few fractures of the tibial plateau or distal femur would be treated operatively without a CT scan - and in Europe many patella cases appear to be treated without plain films so please review the 2nd pargaraph of your introduction. I don't think that there is a lot of misdiagnosis in interpreting radiographs of fractures - there might be a misjudgement of the severity of the fracture pattern

Please explain on line 63 how the random images were selected.

Line 66 - When you say projections do you means the actual images?

Line 70 - Are you stating that patients with the same fracture were seen at different time points - were all your images not initial injury films - there should have been no repeat patients for the same fracture; please clarify this.

Line 72 - who made this decision

Line 85 - what is ESM

Line 89 - same comment as line 70

Line 93 -I just want to confirm that two surgeons manually classified 600 fractures - this case number continues to confuse me based on the samples in your tables

Line 10 1- when you say label d do you mean graded - I am not objecting to the word I just want to be certain I understand?

Line 152 etc - please put the sample size in the text for each overall group of fractures

My confusion over your actual samples continues onto your tables -- please be clear as to the actual number of fractures seen - what the surgeons who manually coded cases found and what the machine identified

Did you use the modifiers as well?

I think the concept of machine learning is reasonable - I think your justification for it needs to modified. Perhaps - fracture identification in clinics without orthopedic trained personnel available - replacement of the need for virtual reading of films in the middle of the night by on-call personnel etc

Reviewer #2: Dear author

This is an interesting article in an area that will only become more relevant. It is well structures and written

I do think that some of the paper needs some simplification to make it more readable to understand the true use of this technology.

I have a couple of other comments

1. How were the 600 tests radiographs actually chosen. I am concerned there may have been some bias in the choice. Why was 600 chosen

2. Were the surgeons involved in reviewing the radiographs part of the design team.

3. I would like to see clearer documentation of how bias was addressed

4. I would like more information on how many fractures out of this selection were not identified compared to the radiologists report. This has a large bearing on the use of this.

Reviewer #3: This paper describes and evaluates a method for knee fracture classification from plain radiographs using a deep neural network. The authors collected around 6000 radiographs from a single hospital and manually classified knee fractures according to the 2018 AO/OTA classification system, including a few custom categories. The authors then trained a simple neural network classifier using a majority of the data and tested its perform on 600 images that were annotated by two experienced observers.

This is overall an interesting study, and the authors especially put a lot of effort into data collection and annotation. The study is focused on the validation of an automatic classification method. A lot of details of this method are missing, which would be logical for a validation study of a method described in detail in another (already published) paper, but this seems not to be the case. For this reason, details of the neural network classifier and its training process should in my opinion be included in the manuscript, preferably in the main text rather than a supplement.

For example, the output of the network is not clear. Does the network generate softmax predictions for X classes? In the supplementary material, the network is described with four filters in the last layer, but here it is not clear how these values are translated into AO/OTA categories. It is also unclear what it means that all images were processed individually by the core section of the network – does this refer to different images from the same patient? The supplement also mentions another dataset used for training, which needs to be mentioned in the main text. The training process should also be described in more detail, at least summarizing how radiological reports were fed to the network in teacher student sessions, and how the other more complex regularization techniques were used such as the autoencoder. The active learning part is also described only very briefly.

For data selection, a random subset of the available images was selected based on the likelihood that the image contained a fracture. How was this likelihood determined and how exactly was it used to select images?

How was the data split into test, training and validation sets – randomly?

The test data was annotated by two orthopedic surgeons. Please give their initials if they are co-authors. I assume that Cohen’s kappa was computed with the readings before the consensus session, it might be worth stating this explicitly.

The values in the result section would be more informative with corresponding confidence intervals.

Minor comments:

- Page 3, Line 41: This sentence is not well written and hard to read: “deep learning; a branch of machine learning, utilizing neural networks; a form of artificial intelligence”

- Page 7, Line 123: “>0.7” should be “<0.7”

- Page 9, Line 175: “seemed to correspond somewhat” is not very precise language

- Page 9, Line 184: “falter” should probably be “failed”

- The resolution of Figure 3 is very low.

Reviewer #4: This is an interesting paper looking at machine learning and its ability to recognize knee fractures. This study is important as it is the start of what will probably become the accepted way of assessing and classifying fractures. It will also allow for the appropriate classification of fractures in an unbiased format allowing classifications to be correlated with results and ultimately to treatment decisions and outcomes. The methodology and statistical evaluation are acceptable. The results are definitely encouraging showing reasonable;le correlation with the AO/OTA classification based only on plain radiographs. The discussion was honest and dealt with the shortcoming and strengths of the research.

6. PLOS authors have the option to publish the peer review history of their article (what does this mean?). If published, this will include your full peer review and any attached files.

Reviewer #1: No

Reviewer #2: No

Reviewer #3: No

Reviewer #4: **Yes: **James F Kellam

---

## [Author Response · Author response to Decision Letter 0]

14 Jan 2021

__See the below word-file for reviewers - it should be more readable than this section (the content is identical)__

Reviewer #1:

 “This is an interesting article but requires some significant clarifications.”

Answer: Thank you and we understand that it is perhaps somewhat unclear; blending orthopedic and machine learning research into a readable format is challenging. We have tried to clarify according to your suggestion.

“The knee is a joint, thus it dislocates but does not fracture. Please use a different word or indicate the bones involved.”

Answer: This is correct and we have adjusted to “fractures around the knee joint”. We believe that enumerating “distal femur, patella and proximal tibia” is overly complex. Some readers will also most likely be from the machine learning community and may find it difficult to understand as they will not be familiar with medical terminology.

“Please clarify your sample sizes; I had great trouble sorting this out.”

Answer: We apologize for the confusion. We have tried to clarify in the beginning of the result section. The sample size is actually rather simple – we have a training set of 6000 images that we use for training and tweaking the neural network. Once the training seems optimal, we have tested the model on 600 images that the network has never encountered. This is very different from traditional orthopedic research, what matters is to have a sufficiently large test-set, basically if the model performs well the training set is adequate in size. If we choose a small test-set, e.g. 100 exams there is a risk that results are due to chance and we will most likely not be able to evaluate many of the rare categories.

Action: We have changed the text and included the requested information on the number of radiographs per exam; see the first paragraph in the results.

“Please provide the number of independent patients -with the number of radiographs per patient.”

Answer: We allowed patients to appear more than once if they appeared with at least a span of 90 days. In the test set about 12 % occurred more than once. We believe that reporting radiographs per patient is less important than the number of radiographs per exam as each evaluation of the network is performed per exam. 

Action: We have added this information to the first paragraph in the results section, together with the information from the previous question.

“Did each patient have all the views -did every plateau fracture have an evaluation with a paired AP and Lateral”

Answer: Yes, the majority had also oblique images that allowed us to evaluate whether depressions were located in the posterior, central or anterior portion of the knee.

Action: Added to the second paragraph under “Data sets”: All images contained at least an AP and a lateral view and had to have the knee joint represented.

“I am not fully clear on machine learning technology - were normal x-rays included if the patient had a distal femur fracture?”

Answer: We filtered as few images as possible as we wanted the radiographs to represent a true clinical setting, i.e. we included even poorly taken images as long as the knee joint was included in the image.

Action: See previous action that includes this question.

“I find the fact that you quote that knee replacements only last a decade in 80% of the population fairly unrealistic - please review the literature to be sure your quoted reference reflects the global feeling.”

Answer: The number 80% is for “post-traumatic” arthritis and not regular primary osteoarthritis where we would expect 95% survival rate (according to the Swedish Knee Registry). The number stems from a perhaps somewhat small study (Lunebourg et al) but a much larger study with shorter follow-up (Bala et al) also find that the complication rate is much higher in this group. 

Action: We have tried to clarify this in the introduction: “While regular primary osteoarthritis have a survival rate of at least 95% in a decade, post-traumatic knee replacements have both higher complication rates and survival rates as low as 80% for the same time period”

1. Lunebourg A, Parratte S, Gay A, Ollivier M, Garcia-Parra K, Argenson J-N. Lower function, quality of life, and survival rate after total knee arthroplasty for posttraumatic arthritis than for primary arthritis. Acta Orthop. 2015 Apr;86(2):189–94.

2. Bala A, Penrose CT, Seyler TM, Mather RC, Wellman SS, Bolognesi MP. Outcomes after Total Knee Arthroplasty for post-traumatic arthritis. The Knee. 2015 Dec 1;22(6):630–9.

3. http://www.myknee.se/pdf/SVK_2019_1.0_Eng.pdf

“In the United States very few fractures of the tibial plateau or distal femur would be treated operatively without a CT scan - and in Europe many patella cases appear to be treated without plain films so please review the 2nd paragraph of your introduction. I don't think that there is a lot of misdiagnosis in interpreting radiographs of fractures - there might be a misjudgement of the severity of the fracture pattern”

Answer: We agree that few (read none) of these would be treated operatively without a CT. The paragraph does though state that the “especially during on call hours” as we usually do CT exams the next day and furthermore only if we are considering surgery. Providing a detailed description in the ER could therefore be of use, especially for selecting which fractures require additional CT-scans. As the other reviewers have not objected to this paragraph we will leave it unchanged.

Action: None.

“Please explain on line 63 how the random images were selected.”

Answer: We agree that “likelihood” was perhaps not the best wording. In effect the procedure was rather simple, we looked for text strings such as “there is a fracture” or “there is a depression” and then made sure that a large part of the images were sampled from those with strings that could have been used with fractures. The word likelihood suggests that we used a statistical method and we agree that this is misleading. As the dataset is dominated by regular knee images it would have been impossible otherwise to retrieve enough fractures to identify all the categories.

Action: We have changed the wording and made it clearer that the report text is guiding the selection.

“Line 66 - When you say projections do you means the actual images?”

Answer: Thank you for this note. We have clarified it to “Radiograph projections”. Many are familiar to this as the “view” but we chose to use projection as this generally the more technical term.

Action: Added “Radiograph” before projection.

“Line 70 - Are you stating that patients with the same fracture were seen at different time points - were all your images not initial injury films - there should have been no repeat patients for the same fracture; please clarify this.”

Answer: Thank you for this remark. The assumption was that a patient would appear with a radiograph of a fracture on day 1, perhaps have a follow-up on day 10 to look for displacement, and then do a final follow-up radiograph on day 40. The most important thought was to avoid having the follow-up image and recruiting the same fracture twice. Similarly, we were less interested looking at healed fractures and thus we extended the period to 90 days.

Action: We have clarified this by changing “first” to “initial” in the description.

“Line 72 - who made this decision”

Answer: The reviewer of the image made the decision if they saw open physes. Although we at the orthopedic department don’t see children there are some neonatal images and occasionally some other clinic that has referred a younger patient to Danderyd University Hospital. While it would be interesting to include these we wouldn’t have enough pathology to find anything useful and immature bone has a different classification.

Action: Added “by the reviewer”.

“Line 85 - what is ESM”

Answer: The extended supplement material

Action: Changed to “supplement”

“Line 89 - same comment as line 70”

Answer: The key factor is that a patient should never appear in different data sets. The network will always overfit the training set and thus it is crucial that as little information as possible leak over into the test-set and we wanted to be very explicit about it here.

“Line 93 -I just want to confirm that two surgeons manually classified 600 fractures - this case number continues to confuse me based on the samples in your tables”

Answer: Yes, this is correct. We hope that our previous answers have made it clearer.

Action: None

“Line 10 1- when you say label d do you mean graded - I am not objecting to the word I just want to be certain I understand?”

Answer: Yes, graded is in our mind a slightly narrower term. The number of classes that the reviewer could choose from was fairly large and not all classes were concerned with the AO classification, e.g. “previous fracture”.

Action: None

“Line 152 etc - please put the sample size in the text for each overall group of fractures”

Answer: Excellent suggestion.

Action: Added the training & evaluation (test) cases to the headers for each fracture group.

“My confusion over your actual samples continues onto your tables -- please be clear as to the actual number of fractures seen - what the surgeons who manually coded cases found and what the machine identified”

Answer: We apologize for this confusion. The machine provides us with a likelihood for a fracture where the simplest measure for how many the machine identified would be to use below or above 50% probability. As we have trained the network with weights proportional to each category’s’ prevalence it will though have a bias towards estimating a fracture. The only number that we look at in practice is the AUC as it is probably the most informative of performance in a clinical setting. When we have tried our network in a clinical setting it is also beneficial that the network is more likely guessing that there is a fracture than not as we are primarily concerned with missing fractures.

Action: We have added a clarification in the table description and renamed the cases column to “Observed cases”.

“Did you use the modifiers as well?”

Answer: We have only used “Displaced” although we named it “Dislocation” which was a mistake on our part (the word “dislocerad” in Swedish means displaced and hence we forgot to properly translate it). Some of the modifiers are not that obvious, e.g. poor bone quality, and combined with the fact that the number of choices in our already rather rich set of classes would have been overwhelming so we decided to focus more on the classes and qualifiers. It is something we will most likely add in some form as we move forward with the project.

Action: We have changed “Dislocated” to “Displaced”

“I think the concept of machine learning is reasonable - I think your justification for it needs to modified. Perhaps - fracture identification in clinics without orthopedic trained personnel available - replacement of the need for virtual reading of films in the middle of the night by on-call personnel etc”

Answer: Yes, we agree that it is suboptimal. The paradox is though in our hospital that while we have no real shortage of radiologist, we have a hard time finding radiologists that are truly interested in orthopedics. This translates into that we usually receive full descriptions of fractures but it unfortunately rarely contains the level of detail that we need to guide our treatments. This is in the end up to the operating orthopedic surgeon and highly influenced by the CT, but we think that there is a missed opportunity of improving the initial report as virtually all fractures have an initial radiograph taken prior to their CT-scan.

Action: We changed the “during on call hours” to “in the middle of the night” and we also added orthopedic expertise to the radiologist.

Reviewer #2: 

“Dear author, this is an interesting article in an area that will only become more relevant. It is well structures and written. I do think that some of the paper needs some simplification to make it more readable to understand the true use of this technology.”

Answer: Thank you. We appreciate that you have taken the time to help us improve the paper.

“I have a couple of other comments

1. How were the 600 tests radiographs actually chosen? I am concerned there may have been some bias in the choice. Why was 600 chosen”

Answer: When we had the 600 tests we felt that we had a decent amount of fractures represented in each category. The difficulty is that in a regular patient flow, most patients will not have a fracture and differentiating between fracture and no fracture was not our objective. As always, more could be better but we believed that 600 would be enough to provide a decent overview of the performance while also putting a reasonable work load as the reviewing process for the test set is much more expensive than the one for the training set. We introduced a bias by design as we have actively looked for reports suggesting fractures as described in our reply to reviewer #1. If we would have filled our dataset with more healthy individuals the results could possibly have been better but we strongly believe that that task would have been too trivial and not of clinical interest. Our goal was to be able to distinguish between severe osteoarthritis and depression fracture, just as we would have to do in a regular clinical setting.

Action: None

“2. Were the surgeons involved in reviewing the radiographs part of the design team.”

Answer: If you refer to the design team then MG all the steps in the study, as well as the reviewing of the radiographs.

Action: None

“3. I would like to see clearer documentation of how bias was addressed”

Answer: In these experimental machine learning studies the bias differs from traditional epidemiological bias. We have tried to not to cherry pick images and excluded only a minimal amount of exams, mostly due to immature bone. The test set with all the 600 images will be made available upon publication so that anyone can evaluate it for sources of bias. The training set is of interest more as an indicator of the amount of data required than an actual source of bias, the bias in the results is all about how obvious the examples are. The more cases in the region of uncertainty, the more difficult it will be for the network to decide on the actual class. This bias is described in the inter-rater table – where a high Kappa-value suggests that we are evaluating obvious cases.

Action: We have added a section in the discussion that discusses these issues in more detail.

4. I would like more information on how many fractures out of this selection were not identified compared to the radiologists report. This has a large bearing on the use of this.

Answer: The fractures reached and AUC of 0.90, we don’t have the true number of reports suggesting fracture but the number should be similar. Our paper from 2017 looked at the detection of fractures and it is certainly an interesting topic but after testing this in a clinical setting, we have found that the true utility comes from the detailed description that we can get from helping clinicians to use complex classifying systems such as the AO/OTA. The difference is equivalent to seeing that there is a car and knowing what car model it is, the former is rather obvious even without an AI while the latter is less obvious but absolutely crucial if you want to order spare parts or do something more interesting. We have skipped this information we wanted to have the paper focused on the AO/OTA classification with minimal distraction. 

Action: We have added the numbers for the fracture class to the supplement together with example images of detection failures.

Reviewer #3:

 “This paper describes and evaluates a method for knee fracture classification from plain radiographs using a deep neural network. The authors collected around 6000 radiographs from a single hospital and manually classified knee fractures according to the 2018 AO/OTA classification system, including a few custom categories. The authors then trained a simple neural network classifier using a majority of the data and tested its perform on 600 images that were annotated by two experienced observers.

This is overall an interesting study, and the authors especially put a lot of effort into data collection and annotation.” 

Answer: Thank you for taking your time to read and grasp the essence of the paper.

Action: None

“The study is focused on the validation of an automatic classification method. A lot of details of this method are missing, which would be logical for a validation study of a method described in detail in another (already published) paper, but this seems not to be the case. For this reason, details of the neural network classifier and its training process should in my opinion be included in the manuscript, preferably in the main text rather than a supplement.

For example, the output of the network is not clear. Does the network generate softmax predictions for X classes?”

Answer: The classes are binary, we predict probability through a sigmoid function of the neuron’s activities. Which is common for these tasks.

Action: None

“In the supplementary material, the network is described with four filters in the last layer, but here it is not clear how these values are translated into AO/OTA categories.“

Answer: The layer’s dimensions are intentionally kept low to make sure that the label information is represented in the shared model’s representation space and it is not overfitting in the fully connected layer’s parameters.

We did not perform an extra experiment on the size of the last layers as it is 

1. expensive and time-consuming for us, 

2. the purpose of the publication is not to find the best architecture for our dataset, but rather, showing the power of deep learning in orthopedical tasks.

3. Even in highly cited publications where the point of the publication is to report the best network architecture and have access to infinite processing power, the choice of minor hyperparameters are generally left to the hunch of the authors. [1,2,3,4]

[1] Krizhevsky, A., Sutskever, I., & Hinton, G. E. (2017). Imagenet classification with deep convolutional neural networks. Communications of the ACM, 60(6), 84-90.

[2] Sermanet, Pierre, et al. "Overfeat: Integrated recognition, localization and detection using convolutional networks." arXiv preprint arXiv:1312.6229 (2013).

[3] Szegedy, Christian, et al. "Inception-v4, inception-resnet and the impact of residual connections on learning." arXiv preprint arXiv:1602.07261 (2016).

[4] He, Kaiming, et al. "Deep residual learning for image recognition." Proceedings of the IEEE conference on computer vision and pattern recognition. 2016.

Action: None

“It is also unclear what it means that all images were processed individually by the core section of the network – does this refer to different images from the same patient?”

Answer: Yes, that is correct.

Action: None

“The supplement also mentions another dataset used for training, which needs to be mentioned in the main text.”

Answer: We recently published AO-classification on ankle fractures and this dataset and other manually-labeled data was used to augment the training set. Our goal is to one day to be able to classify all major bones using AO/OTA. The additional data has some regularizing effects but interestingly we haven’t seen a noticeable improvement for ankles as the dataset for knees grew in size. 

Action: We have added more information on the additional datasets that we used.

“The training process should also be described in more detail, at least summarizing how radiological reports were fed to the network in teacher student sessions, and how the other more complex regularization techniques were used such as the autoencoder.”

Answer: Autoencoders are commonly used for regularization and semi-supervised learning [5]. Our approach is not in any way novel [5,6,7] and therefore we wanted to focus on the orthopedic perspective less than the technical. Those interested in the technical details will have the code available (we will open source the version used in this paper) and the supplement should have the most important details summarized for an overview.

[5] Myronenko, Andriy. "3D MRI brain tumor segmentation using autoencoder regularization." International MICCAI Brainlesion Workshop. Springer, Cham, 2018.

[6] Kunin, Daniel, et al. "Loss landscapes of regularized linear autoencoders." arXiv preprint arXiv:1901.08168 (2019).

[7] Hinton, Geoffrey E., and Richard S. Zemel. "Autoencoders, minimum description length and Helmholtz free energy." Advances in neural information processing systems. 1994.

Action: We have added more details regarding the network to the paper.

“The active learning part is also described only very briefly.”

Answer: Early on we focused on attaining enough representative exams for each category. Once that was done we used the entropy measurement to rank samples for annotation. The latter is what is commonly viewed as active learning although both were part of our targeted sampling procedure. During the process we selected the classes which were at the time most poorly performing for annotation. The selection of cases was made in batches depending on how long it took to annotate the images.

Action: We added the information about targeting the categories to the section for clarity.

“For data selection, a random subset of the available images was selected based on the likelihood that the image contained a fracture. How was this likelihood determined and how exactly was it used to select images?”

Answer: This was a bad use of wording regarding the reports. As we progressed we let the network pick evaluate a new set of 1000-3000 radiographs from which we selected a subset of exams. Early on we wanted to populate all the categories and we selected exams with high probability of belonging to the categories. Later we switched to more traditional active learning where we chose examples in the boundary between the classes of interest where the class was selected based on the performance on the validation set. We have tried to explain it in the methods section but it is difficult to balance the level of detail with readability. 

Action: See above change regarding the report and also the added details on the active learning. 

“How was the data split into test, training and validation sets – randomly?”

Answer: The entire database including the augmentation data was split into the three sets at the beginning based on patients. This was due to concern of patients appearing in multiple sets and thus cause the expected overfitting on the training set to leak over into the test set.

Action: We have tried to clarify in the first paragraph under “Patients & methods > Data sets”

“The test data was annotated by two orthopedic surgeons. Please give their initials if they are co-authors. I assume that Cohen’s kappa was computed with the readings before the consensus session, it might be worth stating this explicitly.”

Answer: Intitials of the two orthopedic surgeons should be given, this was an oversight. 

Action: Have added the initials of the orthopaedic surgeons MG, OS and EA 

“The values in the result section would be more informative with corresponding confidence intervals.”

Answer: We agree, we have added bootstrapped confidence intervals. The idea with the weighting is to be able to provide some sort of summary value that is not too dependent on individual categories, especially the small categories that may just be happenstance.

Action: The confidence intervals

“Minor comments:

- Page 3, Line 41: This sentence is not well written and hard to read: “deep learning; a branch of machine learning, utilizing neural networks; a form of artificial intelligence””

Answer: We agree, we have tried to modify the text for clarity.

Action: Manuscript has been modified.

“- Page 7, Line 123: “>0.7” should be “<0.7””

Answer: Thanks.

Action: Fixed.

“- Page 9, Line 175: “seemed to correspond somewhat” is not very precise language”

Answer: We agree that this is not the most price language and we have tried to modify the text for clarity. The problem is that if we write “no clear correlation” readers will lose important information that the noise in the classes also make them difficult to learn for the network. As the other reviewers haven’t objected to this vague wording we hope that you will find the change sufficient.

Action: Changed to ”appeared to correspond weakly”

“- Page 9, Line 184: “falter” should probably be “failed””

Answer: Yes, fail is a better choice of wording.

Action: Changed

- The resolution of Figure 3 is very low.

Answer: Strange, our resolution is 9730 x 3272 for all figures. The gradients is though low-res as it corresponds to downscaled input image.

Action: We will look into it at resubmission. 

Reviewer #4:

“This is an interesting paper looking at machine learning and its ability to recognize knee fractures. This study is important as it is the start of what will probably become the accepted way of assessing and classifying fractures. It will also allow for the appropriate classification of fractures in an unbiased format allowing classifications to be correlated with results and ultimately to treatment decisions and outcomes. The methodology and statistical evaluation are acceptable. The results are definitely encouraging showing reasonable correlation with the AO/OTA classification based only on plain radiographs. The discussion was honest and dealt with the shortcoming and strengths of the research.”

Answer: Thank you very much for taking the time to review our paper and your kind words. Classification is certainly a subjective task and it is our firm belief that the orthopedic community would have much to gain from an unbiased classification. Too many of our discussions are concerned if a certain study actually included or not fractures that correspond to the discussed fracture.

---

## [Decision Letter · Decision Letter 1]

12 Feb 2021

PONE-D-20-26924R1

Artificial intelligence for the classification of fractures around the knee in adults according to the 2018 AO/OTA classification

PLOS ONE

Dear Dr. Gordon,

Thank you for submitting your manuscript to PLOS ONE. After careful consideration, we feel that it has merit but does not fully meet PLOS ONE’s publication criteria as it currently stands. Therefore, we invite you to submit a revised version of the manuscript that addresses the points raised during the review process.

Both reviewers appreciate the improvements made during the revision. Nevertheless, Reviewer 3 identified several major concerns that I agree with. The manuscript is therefore, not yet ready for publication. Please carefully look at all comments, especially those provided by Reviewer 3 and make sure that the description of the method allows its reimplementation.

We look forward to receiving your revised manuscript.

Kind regards,

Ivana Isgum

Academic Editor

PLOS ONE

Reviewers' comments:

Reviewer's Responses to Questions

**Comments to the Author**

1. If the authors have adequately addressed your comments raised in a previous round of review and you feel that this manuscript is now acceptable for publication, you may indicate that here to bypass the “Comments to the Author” section, enter your conflict of interest statement in the “Confidential to Editor” section, and submit your "Accept" recommendation.

Reviewer #1: All comments have been addressed

Reviewer #3: (No Response)

2. Is the manuscript technically sound, and do the data support the conclusions?

Reviewer #1: Yes

Reviewer #3: Yes

3. Has the statistical analysis been performed appropriately and rigorously? 

Reviewer #1: Yes

Reviewer #3: Yes

4. Have the authors made all data underlying the findings in their manuscript fully available?

Reviewer #1: Yes

Reviewer #3: Yes

5. Is the manuscript presented in an intelligible fashion and written in standard English?

Reviewer #1: Yes

Reviewer #3: Yes

6. Review Comments to the Author

Reviewer #1: Thank you for your thorough responses. I have a few follow-up queries

On line 65 perhaps clarify that you did not look at all plain radiographs but rather those around the knee joint.

Consider re-writing the next paragraph so it is clear who excluded images for quality or open physes etc.

Line 102 - there are three sets of initials but only 2 orthopedic surgeons in your text

I am sorry if I missed this but did you report the agreement between your human raters -- how far down into the codes did they go - how many cases needed reconciliation

You have very few distal femur training cases - is this number adequate?

Were all the A cases evaluated for being A so in the patella you actually have 12 A's , 10 A1 etc

Thank you for these clarifications on your interesting work

Reviewer #3: The revision has improved this paper in several aspects. However, the method (neural network) is still not sufficiently described in my opinion. The argument that the code will be published on acceptance does not appeal to me - the reader should not need to dig through your code to understand your method.

I would strongly recommend moving most of the neural network description from the supplement into the main body of the manuscript, and extending this description. The reader should be able to reimplement the method based on the description in the paper, but there are currently too many details missing.

The “Neural network setup” section needs to contain more details about the whole setup, most importantly how the individual regularization techniques were used together (the reference for using auto-encoders for regularization [21] also refers to a paper about Stochastic Weight Averaging, this seems to be a mistake).

The outputs of the network appear to be sigmoid units, but the table in the supplement lists only 4 output units - it is still not clear to me how these are translated into the different categories. Or is there another layer with the final output units? With sigmoid outputs, two classifications that exclude each other could both have the same probability (e.g. 1.0) - what would be the final classification in such a case?

“The test set consisted of 600 cases, which were classified by two senior orthopedic surgeons, MG, OS and EA, working independently.” - this seems to list the initials of three observers?

The data availability statement has been considerably improved, now the test set will be released. The authors could consider also uploading their evaluation pipeline and their results to a platform like grand-challenge.org where other researchers can then later upload their own predictions for comparison.

7. PLOS authors have the option to publish the peer review history of their article (what does this mean?). If published, this will include your full peer review and any attached files.

Reviewer #1: No

Reviewer #3: No

---

## [Author Response · Author response to Decision Letter 1]

17 Feb 2021

See attached word file for response

---

## [Decision Letter · Decision Letter 2]

8 Mar 2021

Artificial intelligence for the classification of fractures around the knee in adults according to the 2018 AO/OTA classification system

PONE-D-20-26924R2

Dear Dr. Gordon,

We’re pleased to inform you that your manuscript has been judged scientifically suitable for publication and will be formally accepted for publication once it meets all outstanding technical requirements.

Kind regards,

Ivana Isgum

Academic Editor

PLOS ONE

Additional Editor Comments (optional):

The authors have addressed all issues raised by the reviewers and therefore, the manuscript can be accepted for publication. The authors will share the data as soon as PLOS One provides them with a DOI.

Reviewers' comments:

Reviewer's Responses to Questions

**Comments to the Author**

1. If the authors have adequately addressed your comments raised in a previous round of review and you feel that this manuscript is now acceptable for publication, you may indicate that here to bypass the “Comments to the Author” section, enter your conflict of interest statement in the “Confidential to Editor” section, and submit your "Accept" recommendation.

Reviewer #1: All comments have been addressed

Reviewer #3: All comments have been addressed

2. Is the manuscript technically sound, and do the data support the conclusions?

Reviewer #1: (No Response)

Reviewer #3: Yes

3. Has the statistical analysis been performed appropriately and rigorously? 

Reviewer #1: (No Response)

Reviewer #3: Yes

4. Have the authors made all data underlying the findings in their manuscript fully available?

Reviewer #1: (No Response)

Reviewer #3: Yes

5. Is the manuscript presented in an intelligible fashion and written in standard English?

Reviewer #1: (No Response)

Reviewer #3: Yes

6. Review Comments to the Author

Reviewer #1: (No Response)

Reviewer #3: All comments have been addressed and the manuscript has been considerably improved. No further concerns.

7. PLOS authors have the option to publish the peer review history of their article (what does this mean?). If published, this will include your full peer review and any attached files.

Reviewer #1: No

Reviewer #3: No

---

## [Editor Report · Acceptance letter]

22 Mar 2021

PONE-D-20-26924R2 

Artificial intelligence for the classification of fractures around the knee in adults according to the 2018 AO/OTA classification system 

Dear Dr. Gordon:

I'm pleased to inform you that your manuscript has been deemed suitable for publication in PLOS ONE. Congratulations! Your manuscript is now with our production department. 

Kind regards, 

on behalf of

Professor Ivana Isgum 

Academic Editor

PLOS ONE